# Creating Strong Titanium/Titanium Hydride Brown Bodies at Ambient Pressure and Moderate Temperatures

**DOI:** 10.3390/ma13215008

**Published:** 2020-11-06

**Authors:** Jonathan Phillips, Anthony Janssen, Troy Y. Ansell, Claudia C. Luhrs

**Affiliations:** 1Energy Academic Group, Naval Postgraduate School, Monterey, CA 93943, USA; 2Mechanical and Aerospace Engineering, Naval Postgraduate School, Monterey, CA 93943, USA; tjanssen@gmail.com (A.J.); troy.ansell@nps.edu (T.Y.A.); ccluhrs@nps.edu (C.C.L.)

**Keywords:** hydrogen-enhanced atomic transport, titanium fabrication, metal sintering

## Abstract

A simple, low temperature, method, hydrogen-enhanced atomic transport (HEAT), for creating metallic-bonded brown bodies of order 40% bulk density in molds of designed shape from Ti metal particles is introduced. In this initial study 40 micron titanium particles were poured into graphite molds, then heated to temperatures equal to or greater than 650 °C for four hours in a flowing ambient pressure gas mixture containing some hydrogen led to brown body formation that closely mimicked the mold shape. The brown bodies were shown to be dense, metallic bonded, and consisted of primarily Ti metal, but also some TiH. It is postulated that hydrogen is key to the sintering mechanism: it enables the formation of short-lived TiHx species, volatile at the temperatures employed, that lead to sintering via an Ostwald Ripening mechanism. Data consistent with this postulate include findings that brown bodies are formed with hydrogen present (HEAT process) had mechanical robustness and only suffered plastic deformation at high pressure (ca. 5000 Atm). In contrast, brown bodies made in identical conditions, except the flowing gas did not contain hydrogen, were brittle, and broke into micron scale particles under much lower pressure. HEAT appears to have advantages relative to existing titanium metal part manufacturing methods such as powder injection molding that require many more steps, particularly debinding, and other methods, such as laser sintering, that are slower, require very expensive hardware and expert operation.

## 1. Introduction

In this paper a novel method for manufacturing brown titanium/titanium hydride parts (open pore metallic foam) from titanium particles, a method mechanistically related to reduction expansion synthesis-sintered metal (RES-SM), but based on a unique chemistry is described. It is potentially an alternative to standard additive manufacturing processes that use laser induced melting. It is also a potential alternative to metal injection molding (MIM), as MIM processes require more steps and higher temperatures to create brown bodies similar to those resulting from the lower temperature process introduced here. Indeed, any new process that is shown to reduce the required processing temperature by several hundred degrees, as shown herein, is significant.

The new process, hydrogen-enhanced metal transport (HEAT), is simple in practice: at elevated temperatures and ambient pressure a gas mixture containing hydrogen and inert gas is flowed over a bed of only titanium particles, held in an inert (e.g., graphitic) mold. At moderate temperatures, where moderate means more than 500 °C below the melting temperature of the metal processed, and with sufficient time (ca. 4 h) brown bodies of shapes that closely follow the contours of the original molds form. These brown bodies are composed of two phases: titanium and titanium hydride.

It is postulated that titanium/titanium hydride metal bodies using HEAT occurs via this mechanism: titanium hydride, TiH_2_ readily forms in a hydrogen-containing environment as predicted by thermodynamic phase diagrams. It is anticipated that, as per kinetic theory, the titanium hydride concentration reaches steady state because of a balance between formation and decomposition. It is postulated that some TiHx species have a relatively high vapor pressure at temperatures just below or order of 200 °C above the decomposition temperature of the hydride, leading to “transport” of this species in molecular form within the bed. Subsequent decomposition leads to titanium deposition/transport. As shown below, this classic model of sintering is consistent with all observations.

The proposed model of sintering in the HEAT process is that it is a form of Ostwald Ripening (OR), and that Ostwald Ripening does not occur in the absence of hydrogen gas. Thus, the nature of titanium sintering observed in the present study is unique. The general OR model is that transport of material within a particle bed, either surface or gas phase, at the atomic/molecular scale leads to large particles growth at the expense of small particles [1,2]. A similar “metal radical” model is used to explain the growth of large metal particles in the gas phase during catalytic etching [3] as well as the enhanced rates of particle growth in supported catalysts under reaction conditions [4]. In the case of the beds to titanium particles treated in this study, the process is postulated to lead to growth in the average particle size via sintering and neck formations between particles. The ultimate product is a titanium brown body of moderate density. All experimental evidence, as detailed below, is consistent with this postulated mechanism.

The development of the HEAT process represents a further evolution of reduction expansion synthesis (RES) [5,6,7,8,9,10,11,12,13,14,15,16], a set of technologies based on a “reduction” chemistry introduced by our team. RES chemistry in all cases starts with a primary step: thermal decomposition of a “reducing” solid, such as urea, under inert gas. This primary step chemistry creates volatile reducing species that can be harnessed to create a wide range of products based on designed secondary reactions. Processes developed on the basis of the RES concept include the batch generation of sub-micron metal and metal alloy particles [7,8,9], metal thin film formation [13,14], metal part formation from mixtures of metal and metal oxide particles [6,15,16], graphene from graphite oxide [10], even stable tin/carbon anodes for batteries [11].

In all “metal” variants of RES [6,7,8,9,13,14,15,16] there is also a key secondary step: radicals, produced in the first step, interact with oxygen atoms in metal oxides or hydroxides to create products such as CO_2_ and H_2_O that subsequently leave the process reactor as a gas. The heavier metal atoms/metallic clusters produced via the removal of oxygen are not volatile and do not leave the reactor. Instead, based on proper arrangement of materials within the bed, these species migrate, as per the OR mechanism, leading to metal particle growth and sintering. The process has been demonstrated to lead to growth and sintering of metal, specifically Ni, Fe, and Cr, objects from beds or films consisting originally of particles of metal oxide and in some cases metal. This leads to the creation of designed solid metal objects by the addition of multiple layers, thus this type of RES, RES-sintered metals (RES-SM) is an alternative to standard metal additive manufacturing or even particle injection molding.

The inherent chemical limitations of the RES-SM method to only a few metals led to the invention of HEAT. Indeed, RES will not work for certain metal oxides, including titania, which are too stable for reduction by radicals even at 1000 °C. Yet, the motivation for finding an RES-like process for metal additive manufacturing of titanium parts is clear. Titanium and titanium alloys are widely used in aerospace, automotive, chemical, and biomedical industry because of its great strength, low weight, and excellent corrosion resistance [17,18,19]. Also, titanium is very expensive, thus “subtractive” manufacture involves a significant cost, because cut metal must be reprocessed. Thus, HEAT was developed as a means to create a new “primary step” for titanium sintering.

The two main steps in the postulated HEAT process parallel those in the RES-SM. The generation of mobile, but short-lived, titanium hydride species is postulated to be the primary step in HEAT. This is equivalent to the generation of mobile atomic species via local oxide reduction in the RES-SM process. The subsequent HEAT secondary steps lead to OR, very similar to that in RES. In HEAT and RES-SM, the dominant means of metal transport is hypothesized to be via atomic metal species, which leads, via OR, to sintering. HEAT, like RES-SM, creates brown bodies at ambient pressure and at a temperature far below the melting temperature of the metal.

Prior to the present study, mold-based methods for titanium part creation were the MIM process [20,21], and high pressure cold pressing as RES cannot be used to create titanium brown bodies. Many variations on the MIM process have been studied, including the use of titanium hydride as a “sintering aid” [22,23,24,25]; however, in all cases both high temperature (>1000 °C) and high pressure, more than 1000 atmospheres are required to create brown bodies demonstrated to have high strength. Also, the MIM process is multi-step: (i) A green body containing a binder (wax) is formed by high pressure injection into a metal mold followed by moderate heating. (ii) The binder is removed in a slow heating step (debinding). (iii) The metal is sintered at high temperature to create a titanium brown body. An advantage to the HEAT process, similar to the advantages of RES, is that neither high temperature nor pressure is required to create a strong titanium brown body of designed shape. Only one step is required: heat (>650 °C) titanium particles in a mold at ambient pressure in a gas containing hydrogen for a few hours.

In sum, this study provides evidence that sintering Ti in a hydrogen-containing gas allows for the formation of hard brown bodies of Ti at 650 °C which are structurally equivalent to those produced in inert gas at about 1000 °C. This may provide a way forward to create strong brown, designed, titanium and titanium alloy metal parts more quickly and with less investment than any commercial process for creating titanium parts of designed shape from particles including laser particle bed sintering and related metal additive manufacturing technologies, and metal powder injection molding (MIM).

## 2. Materials and Methods

### 2.1. Sample Preparation

There are three parts to the HEAT process employed herein. First, for each experiment a new mold was created from Grafoil (Union Carbide, Lakewood, OH, USA). Second, titanium metal particles were added, simply by pouring particles from the source and gently brushing to create a flat top surface, until the mold was at approximately half capacity. Third, the mold/titania powder was fired in a simple flush/heat sequence. Each step is described in more detail below.

#### 2.1.1. Mold

The molds were created from Grafoil (GTA Grade 0.3 mm thick, NeoGraf Solutions, Lakewood, OH, USA), a moderate surface area, ~22 m^2^/g, graphite material [26,27] made from compressed graphite flakes. This material can be cut and “shaped” in the same manner as cellulose-based construction paper. Small cylindrical molds approximately 1.0 cm in diameter by 0.5 cm tall were easily created for each experiment (Figure 1). In addition, to add mechanical stability, the molds were placed in a “stand,” also made of Grafoil. The stand consisted of 10 layers of Grafoil formed into a rectangle, approximately 4.5 cm × 1 cm. A hole was punched in the center of the stand to accommodate the mold (Figure 2 and Figure 3).

#### 2.1.2. Experimental Conditions

In all cases reported herein (Table 1) the precursor material was titanium powder (Sigma Aldrich – Ti powder, 325 mesh, 99.9% metal basis), average particle size (~40 microns). The weight of the input titanium powder was in all cases 0.5 +/− 0.1 gms. The key treatment parameters, such as temperatures and gases employed, along with the apparent bonding quality from visual observations for all samples is found in Table 1. The term Strong Brown Body (SBB) is employed here to describe products that do not easily pulverize or break under compressive forces of more than 5000 Atm.

#### 2.1.3. Firing

The firing process required seven simple steps. (i) The mold and stand were placed in the center of a 50 cm × 2.5 cm diameter quartz tube. (ii) The tube was flushed with ultra-high purity (UHP) argon (Praxair, Salinas, CA, USA) or a pre-mix gas of argon and 2% hydrogen (Praxair, Salinas, CA, USA) at ~50 sccm for 30 min. (iii) Gas flow reduced to ~10 sccm. (iv) Tube, sealed with no change in gas flow (per iii) quickly placed inside a furnace (Lindburgh/Blue M, 24” single zone) pre-heated to the test temperature such that the mold/sample was at the furnace center. (v) The quartz tube/sample, gas flow continuous, held at test temperature for four hours. (vi) At the completion of the test time the quartz tube was quickly removed from the furnace and the gas flow increased to 50 sccm. (vii) The quartz tube/sample cooled, under gas flow, at ambient temperature, generally for 30 min. At the end of this process the tube was opened and the sample removed for observation. (A picture of the furnace and the tube is available in Reference 16.)

### 2.2. Characterization and Analysis

Four characterization techniques were employed to analyze the solid objects created. (i) A Rigaku Mini-flex 600 X-ray diffractometer (Rigaku Corporation, Tokyo, Japan) operated at 40 kV and 15 mA with a Cu metal target (1.54 Å Kα line) was used for x-ray diffraction of crystal structure. Data were collected in the 2θ range of 10° to 90° at 3°–5° min^−1^, step width 0.02°. Structural and refinement data analysis were performed using the software Jade 9. (ii) Micron and sub-micron scale morphology of Ti specimens was studied with a Zeiss Neon 40 scanning electron microscope (ZEISS International, Oberkochen, Germany). Key parameter settings were a 30 μm aperture and an accelerating voltage of 20 kV. (iii) An Archimedes principle device, “Ohaus density kit” (Ohaus, Persippany NJ, USA), was employed in an effort to determine density. The Archimedes method results were found to be unrepeatable and clearly suggested an exaggerated density. This is consistent with the general finding that the Archimedes method is only reliable for titanium samples with density acceding 95% of nominal solid density [22,28,29,30]. Thus, like others [28,29,30], for the low density samples made in this study, a digital micrometer with a sensitivity of 0.02 mm was employed. Per standard practice values reported are based on average five measurements of thickness and diameter. (iv) A model 5892 Instron tester (Instron, Norwood, MA USA) was used in ambient temperature compression mode (max capacity 100 kN) to determine density vs. compressive pressure behavior. In this work, the brown body “cylinders” were not confined by any device during compression.

## 3. Results

All data are consistent with the major hypothesis of this study: hydrogen gas passed through a bed of Ti particles enhances sintering because of the formation of unstable, but volatile (at the temperatures of the experiment) hydride species that concomitantly lead to metal transport and sintering/growth of metal via an Ostwald Ripening process. Specifically, it was demonstrated that heating (T > 650 °C) Ti particles at ambient pressure in hydrogen-containing gas leads to the formation of solid brown bodies, with a shape closely matching the graphitic mold shape. In the absence of hydrogen, all other parameters remained unchanged, a solid brown body does not form until ~950 °C.

### 3.1. Visual Observations

Solid cylinders with the same radii as the mold formed above 650 °C for HEAT, and above 850 °C for control processes (no hydrogen). At lower temperatures solid structures appeared to be present, but after even modest shaking only powder was found (Figure 2). The control sample behavior is completely consistent with earlier work that shows parts made from Ti or TiH_2_ powders and sintered below ~950 °C in Ar or vacuum are very fragile [31].

### 3.2. X-Ray Diffraction Phase Analysis

The phases of Ti found after treatment are clearly a function of the gas present during firing (Figure 3). The patterns presented in Figure 3A,D are those of commercially acquired Ti and TiH_2_ powders, respectively. After titanium powder treatment in argon, at all temperatures, only titanium metal is present (Figure 3B,C), and the XRD line positions are identical to those of the precursor particles. In contrast, after heat treatment in Ar/H_2_ 98%/2%, although the patterns are dominated by metal peaks, there is always evidence of titanium hydride phases (Figure 3E,F). For all samples treated in Ar/H_2_ gas, there are diffraction lines unique to TiH_2_, as well as peak widening and evidence in the baseline that the peaks are formed by the contribution of more than one phase. These features are not observed in the samples fired in only Ar.

Employing PDXL (Rigaku proprietary software) to determine phase fractions from integrated line areas yields an estimate, after treatment in forming gas at 650 °C (Test 10), of 30% TiH and 70% Ti metal. After treatment in forming gas at 850 °C this method yields an approximate composition of 10% TiH and 90% Ti metal. Although the absolute values are only semi-quantitative, the method does provide a valid approximate indication that the amount of hydride decreases with firing temperature. This is consistent with thermodynamics indicating TiH_2_ is less stable as temperature increases. In fact, the HEAT sample produced at 900 °C (Table 1 NO. 14) did not yield any TiHx lines in XRD. This is consistent with earlier studies in which no hydride was found after treatment at 1000 °C of pure titanium hydride powder [23].

### 3.3. Microstructural Analysis Using Scanning Electron Microscopy

The SEM images support the proposed model of HEAT; however, the evidence is subtle, for example there is little evidence of extensive ‘neck’ formation between particles. This contrasts with the less subtle SEM evidence, including extensive necking, observed in earlier RES-SM studies [6,15] of solid, designed shapes. Chief among the differences observed between HEAT and control samples is the smoother profile and edges of the particles formed in Ar/H_2_ mixtures, the identification of direction joining of some particles, and the appearance, only for the Ar/H_2_-treated samples, of structures associated with multiple phases.

All of the above listed differences between Ar/H_2_- and Ar-only-treated samples can be observed in Figure 4, and are even more evident in Figure 5. First, the micrographs suggest “rounding” of all particle edges in the HEAT-processed samples. This structure is consistent with prior studies of particles growing via OR type sintering [1,2,3,4]. In those works, the same “rounding” of edges is observed. Also, only for the HEAT-prepared samples is there evidence of chemical connections between particles. The structures observed are not “necks,” but clearly show diffusion between adjacent particles at particle edges. Finally, only in the HEAT samples are “stacking” or lamellae structures observed that are similar to those found in multi-phase titanium alloys [30].

All the features of the SEM results are consistent with the other findings. First, it is clear that there is a great deal of void space, in agreement with density studies (Table 2). Second, the XRD results (Figure 3) clearly indicate that there are multiple phases present in the HEAT samples. Third, following shaking (Figure 2) and compression studies (Figure 6 and Figure 7) only HEAT samples are evidently held together by metallic bonds throughout. The HEAT samples are stable to shaking and only metallic bonds can explain the plastic deformation observed under high pressure. In contrast, the thermally sintered control samples shatter in a brittle fashion under minor shaking or compression (Figure 7).

### 3.4. Density vs. Applied Pressure Tests

In the absence of a mold, compressed HEAT-generated strong brown bodies demonstrate the existence of metallic bonding. As illustrated in Figure 6 the brown bodies sintered at 750 °C were modified in shape by high-pressure compression (approximately 5000 Atm), but only modestly. The diameter of the near-cylinders increased by roughly 2.5%. These modest dimensional changes are indicative of metallic bonding holding particles together throughout the sample. In contrast, weakly bonded bodies (WBB) samples made by sintering in pure argon at 750 °C were found to be brittle, consistent with earlier studies [31]. Even after relatively low loading (ca. 1000 Atm) the samples completely crumbled (Figure 7). This indicates true metallic bonding did not exist in the control samples. 

It is interesting to compare the density versus the compressive force data for the HEAT samples with earlier reports regarding the use of cold compression to create brown bodies from titanium and titanium hydride powders [28,29]. It must be noted in making the comparison, many earlier reports regard titanium hydride, not pure titanium, and that all prior studies confined the powder and compaction product using hard steel molds during compression. Hence, it is anticipated that the earlier outcomes (Figure 8) can only serve as a point of qualitative comparison. In some cases, the difference in protocol between the present work and earlier work can lead to enormous variation in observed behavior. For example, the absence of a mold for the control samples in the present study is likely the cause for the observed brittle behavior.

The outcome of density vs. compressive pressure studies of HEAT-treated samples (Figure 8) follow similar density vs. compression trend lines observed in earlier cold compression studies of both Ti and TiH_2_ particles. The primary difference is that the trend line for HEAT samples fired at 750 °C indicates HEAT-prepared samples appear to have more resistance to compressive forces. At any given compression above ~100 MPa, the cold compressed samples have a density 10% more than that of the HEAT-prepared samples. This difference may partially reflect the difference in constraints. Only the unconstrained HEAT samples can expand orthogonal to the compressive force, leading to increased lateral dimension. As density is mass/volume, the denominator is larger for unconstrained compression. Still, based on observed changes in diameter, this lateral expansion should increase density due to volume increase by <4%, whereas measured density changed by more than 10% in all cases.

The control sample fired at 950 °C appears to lie exactly on the curve of the earlier cold compressed samples. Only one control is shown, because the controls prepared at lower temperature disintegrated during compressive testing. In sum, it is concluded that uncompressed HEAT-prepared samples are approximately as “hard”/resistant to compression as cold-compressed Ti and TiH_2_ powders, and have the advantage of achieving that property without compression.

## 4. Discussion

All empirical evidence shows the HEAT process leads to the formation of metallic-bonded titanium brown bodies of any desired shape (matching the mold). The process simply requires heating titanium particles in a mold at ambient pressure and temperature as low as 650 °C. It also should be noted that only titanium brown bodies generated using the HEAT process contain hydride. As all other studies involved sintering at above 1000 °C, all hydride present initially decomposed as expected by thermodynamics [22,23,24,25]. Indeed, the temperatures and pressures required to create HEAT brown bodies are significantly lower than those employed in any prior work, ca. > 1000 °C, and >1000 atmospheres of pressure [19,20,21,22,23,24,25], designed to create titanium brown bodies. The significance of employing hydrogen in the HEAT process is clear: brown bodies generated in the control studies, that is samples prepared identically to the HEAT process below 950 °C, except for the absence of hydrogen gas, are very brittle, and somewhat brittle even if heated to 950 °C.

The data suggest a simple model of the mechanism of metallic-bonded brown body formation; hydrogen gas creates a short-lived volatile titanium compound that mobilizes titanium atoms or clusters. The hydrides decompose, releasing hydrogen gas, and metal atoms or clusters. The Ti species released in this process re-bond with existing particles, and this eventually leads to particle sintering and the creation of metallic “connections” between existing titanium particles in the bed. The process of sintering described is anticipated by one of the oldest models of particle growth: Ostwald ripening [1,2,3,4].

All observations are consistent with this model. First, visual inspections and simple “shaking” tests reveal that only brown bodies produced in the presence of hydrogen at temperatures greater than 650 °C are mechanically stable. In contrast, the control samples prepared at 850 °C or less are clearly very brittle, even under low loads. Second, XRD studies reveal that some titanium hydride is present in HEAT-prepared samples, but not in the control samples. This finding is anticipated by/consistent with the model. Note: Another fact consistent with the model: the Ti/TiH_2_ phase diagram [32] indicates both metal and hydride phases should coexist at the temperatures employed. Third, SEM reveals the existence of features such as softened edges and evidence of direct particle–particle sintering, anticipated for an OR process. The existence of step structures observed in SEM, similar to structures observed in “stabilized multi-phase titanium alloys” [32,33], is consistent with the XRD results showing the coexistence of metal and hydride phases in the HEAT samples. Fourth, the finding that HEAT prepared samples modestly plastically deform under pressure is clearly an indication that these samples have metallic bonds formed between particles. In contrast, the control samples created at 850 °C or lower completely shatter under pressure, revealing a lack of metallic bonding between particles. This clearly supports the supposition that hydrogen is needed to promote diffusion and generation of metallic bonds between particulates.

In addition to the data reported being consistent with the provided model, a number of literature reports are consistent with the postulate that short-lived volatile metal-containing species can lead to gross modification of metals [3]. The earliest reports of atoms/radicals restructuring metals were published more than 90 years ago [34,35,36,37]. Of greatest relevance are reports that hydrogen atoms can restructure metal, which dates from 1933 [38]. It was reported that hydrogen atoms restructured arsenic, antimony, selenium, tellurium, germanium, and tin. The ability of hydrogen atoms to volatize/restructure metals was repeated more recently [39]. In an even more recent report it was found that in the afterglow of a microwave generated hydrogen plasma tin foils were completely reorganized to create a low-density network of micron scale “strings” [40]. Remarkably, detailed study showed the final structure was also tin and the restructuring took place without any weight loss. All evidence suggests the restructuring occurred via metal transport in the form of unstable tin hydride. There are also several reports of the restructuring of platinum because of the action of “radical” species, formed homogenously during the combustion process of both ethylene [3,4,41,42] and hydrogen [43,44] oxidation. Reportedly, these radicals react with platinum foils, films, catalysts etc., to create very short-lived volatile species. Upon decomposition, new platinum structures form. The net result is gross-scale reconstruction over time, at temperature hundreds of degrees below melting, often clearly observed without any magnification.

In all published reports of hydrogen-induced metal reconstruction, the emphasis is on the action of hydrogen atoms, not molecules. Is it possible that hydrogen atoms are the actual active species in HEAT? There is data regarding the H-atoms in titanium. For example, it is known that within titanium hydrides of all stoichiometries, hydrogen atoms are the diffusing species and diffuse rapidly within the crystal even at low temperature [45]. It is also clear there are a multitude of titanium hydride stoichiometries, of formula TiH_2−x_ where x can be greater than one, are known to exit. This clearly indicates that titanium is able to “split” hydrogen molecules. Moreover, it is well-known that platinum, palladium, and other noble metals can split hydrogen at even ambient temperature. Hydrogen atoms created in this fashion are used in catalytic isomerization reactions [46,47,48,49]. It has even been demonstrated that hydrogen atoms created in this fashion can diffuse through the gas phase, impacting processes downstream [50]. In sum, although there is no direct study of atomic diffusion on titanium at elevated temperatures, the suggestion is consistent with observations regarding atomic hydrogen formation and diffusion on other metals.

There are at least two alternative models of hydrogen-induced titanium sintering in the existing literature, which bear consideration, yet it must be noted both are based on sintering studies performed at temperatures and pressures far higher than that employed herein. Also, the models are generally consistent with observation, but contain no direct supporting evidence. The first is that surface oxide is a barrier to sintering, and hydrogen formed via titanium hydride decomposition reacts with surface oxygen to form water, enabling faster, more complete sintering of titanium hydride relative to titanium [22]. In all cases in which titanium hydride was initially present, the final product was pure titanium metal [22,23,24,25]. In all those studies with pure titanium, sintering is conducted at 1200 °C or even higher, that is 550 °C higher than that required to observe sintering in the present study, and in all cases sintering was conducted on cold pressed compacts made at 1000 Atm or higher pressure. Notably, there is no direct evidence provided of a layer of oxide on the particles, nor of its removal. The only evidence provided is the generation of water, at a rate <3% that of hydrogen generation by decomposition [22]. Water is a ubiquitous impurity, hence the source is always uncertain. Moreover, in the one study in which oxygen content was measured, sintering increased, not decreased, oxygen content [25]. The inclusion of TiH*_x_* particles is shown to enhance sintering, presumably acting as a hydrogen source for oxygen removal. In sum, it is not clear those results pertain to the work described herein for which there are no TiH_2_, particles in the initial bed, and sintering is significant even at 650 °C.

The second model is based on sintering in hydrogen-containing gas of mixtures of titanium, aluminum and vanadium particles in a stoichiometry of the target alloy, Ti-6Al-4V, in hydrogen-containing gas [51]. In all cases sintering of high density compacts, created via cold pressing at 3500 Atm., was carried out at 1200 °C and followed by a multi-step cooling protocol including a prolonged anneal in vacuum. It was postulated that solid titanium solutions containing hydrogen have a reduced activation energy to self-diffusion, hence a mechanism for metal transport and consequent sintering, relative to pure Ti. It is not at all obvious that the model, for which there is no direct evidence, can be extrapolated to explain observations made for sintering loose, pure, titanium powder at ~500 °C lower temperature.

## 5. Conclusions

All data indicate that the HEAT process, specifically treating a mold filled with titanium particles to above 650 °C in a flowing ambient pressure gas containing hydrogen, creates a metallic bonded brown body on the order of 40% metal density. The brown body will also mimic the mold shape. Based on prior work with titanium brown bodies formed using Ti-MIM [21] it is anticipated that second step of hot isostatic pressing will lead to complete densification.

It must be noted that finding a low cost, relatively rapid alternative to standard “subtractive” manufacturing is particularly valuable for titanium and titanium alloys. Not only is titanium more expensive than other metals of construction, it is also clear cutting, drilling, and other standard metal shaping processes are particularly difficult for titanium and its alloys [52,53]. It is argued below that HEAT is arguably the most viable alternative to standard titanium manufacturing. To wit: the HEAT process may have advantages relative to current commercial additive manufacturing processes, particularly as it creates brown Ti bodies at 650 °C and ambient pressure, with properties equivalent to those found for cold-pressed samples following conventional sintering processes, that is in Ar or vacuum at ~1000 °C. Lowering the temperature required for any industrial process by several hundred degrees is fundamentally significant. Another advantage to the HEAT process is simplicity. For example, HEAT creates a titanium brown body in a single step in a soft mold. In contrast, Ti-MIM requires a sequence of steps to create a similar brown body: (i) mixing metal and binder, (ii) high pressure injection and high pressure molds, (iii) moderate temperature green body formation, and (iv) very slow debinding at 900 °C. Laser sintering from particles requires extremely expensive equipment, is slow, and requires significant expertise. Finally, all data collected are consistent with a simple model; in the presence of hydrogen at temperatures exceeding approximately 650 °C, some form of titanium hydride is produced. The vapor pressure is sufficient to carry titanium short distances within the bed before decomposing. Titanium atom/clusters generated by the decomposition are released and these bond with existing particle surfaces. The net result, as anticipated by the Ostwald Ripening model, is sintering and metal bonding between particles.

This work was designed for comparison with existing metal “additive manufacturing” technologies, particularly mold-based MIM. In the MIM process, as in true metal additive manufacture by laser sintering, generally a fully dense metal object is the required product. Alternatively, the “brown” Ti parts created here can be considered “designed” open cell Ti/TiH metal foams of high compressive strength. Cellular metal foams, as discussed elsewhere may have properties for some applications superior to fully dense metal parts [54,55]. For example, the light weight, yet high strength (Figure 8), of such foams could be advantageous for applications in aerospace, sports equipment, and as bio-compatible [56,57,58,59,60,61,62] implants. Thus, it is possible that the brown bodies of specific shape produced with the HEAT process may represent an end state for some applications, with no need for further densification.

## Figures and Tables

**Figure 1 materials-13-05008-f001:**
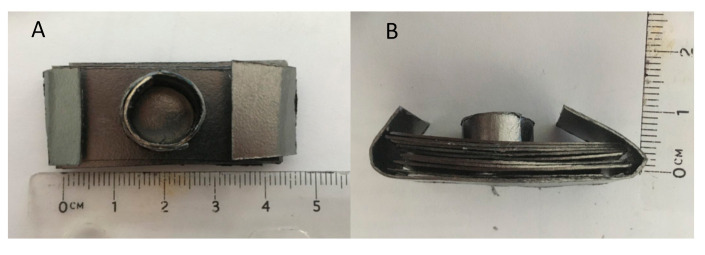
Grafoil mold. (**A**) Top view of handmade mold. (**B**) Side view. Powder is poured into the mold. No compression undertaken.

**Figure 2 materials-13-05008-f002:**
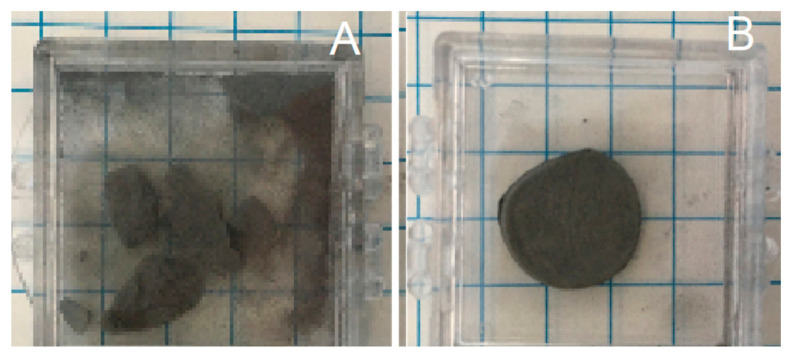
Stability after mild shaking. (**A**) Control sample, produced in Ar only at 750 °C. Sample displays brittle behavior, breaks, and pulverizes. (**B**) Sample, imperfect cylinder shape that mimics mold shape, created in Ar/H_2_, same temperature conditions, shows metallic behavior, remaining intact after shaking. (Scale: each square is ¼ in wide).

**Figure 3 materials-13-05008-f003:**
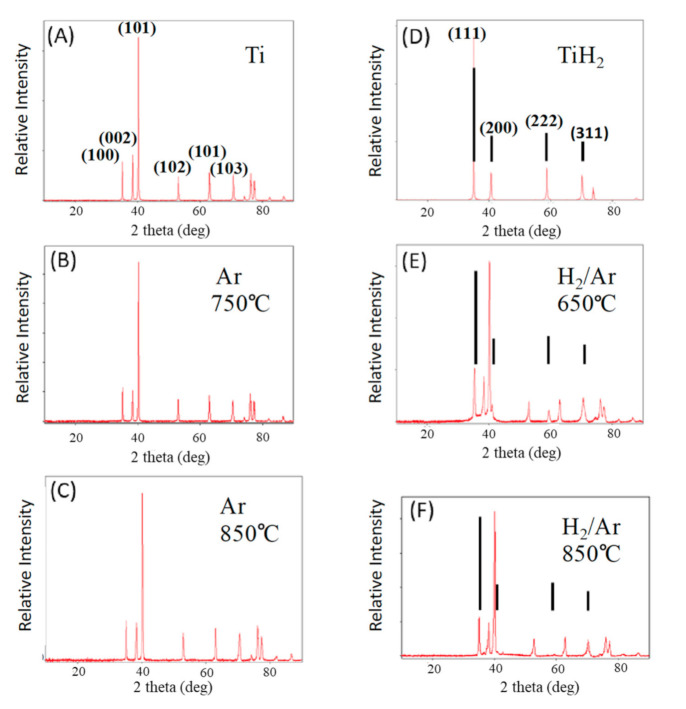
XRD of fired samples—firing temperature and gas phase shown. (**A**) Commercial Ti powder. (**B**,**C**) Pure titanium samples fired in Ar. (**D**) Commercial TiH_2_ powder. (**E**,**F**) Samples fired in forming gas. These clearly contain both Ti metal and TiH_2_. The diffraction lines marked with the line segments are clearly TiH_2_, per D. All unmarked lines are Ti metal lines, as per A.

**Figure 4 materials-13-05008-f004:**
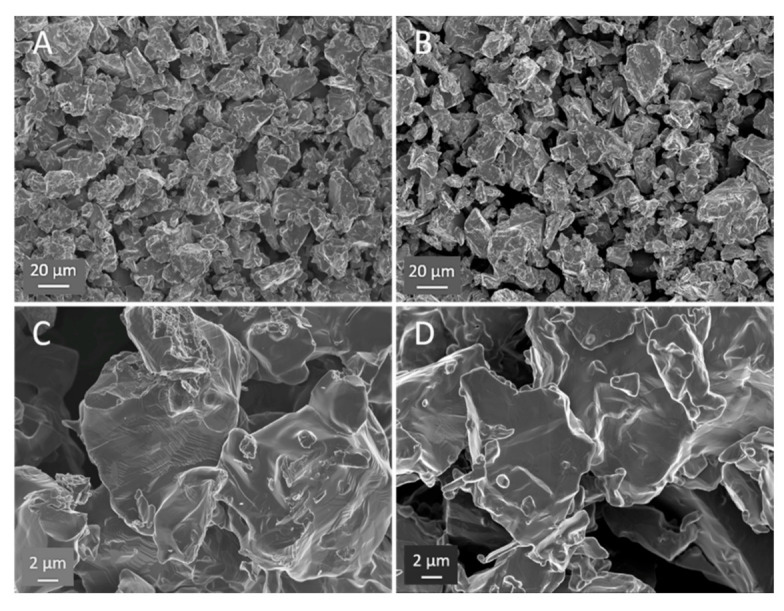
SEM contrast HEAT vs. control. (**A**) and (**C**) HEAT at 850 °C. (**B**) and (**D**) Ar only 850 °C. HEAT samples show evidence of diffusion/metal flow; smoother particle edges, evidence of sintering between particles, and examples of stacked layers (center of (**C**)), characteristic of titanium alloys with multiple phases. Note, the SEM appearance of the Ar-treated control samples appear virtually identical to the angular starting powder.

**Figure 5 materials-13-05008-f005:**
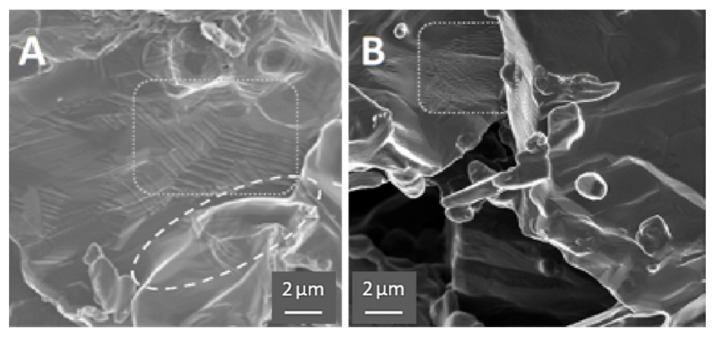
HEAT vs. control. (**A**) Prepared with HEAT and shows evidence of sintering (ellipse area) and “stacked layers” (rectangular area). (**B**) Control sample prepared in pure Ar and shows small wrinkles/crenulations (rectangular area), but no evidence of sintering, stacking, or growth in SEM images.

**Figure 6 materials-13-05008-f006:**
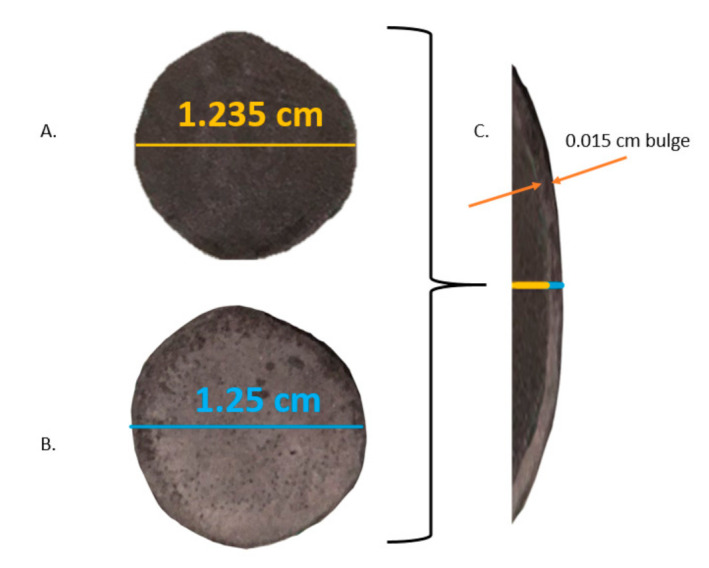
Yield under compression (**A**). HEAT-prepared sample prior to compression (**B**). Same sample following approximately 5000 Atm compression (**C**). Overlay of pre and post compression-diameter of specimen increased by approximately 2.5%.

**Figure 7 materials-13-05008-f007:**
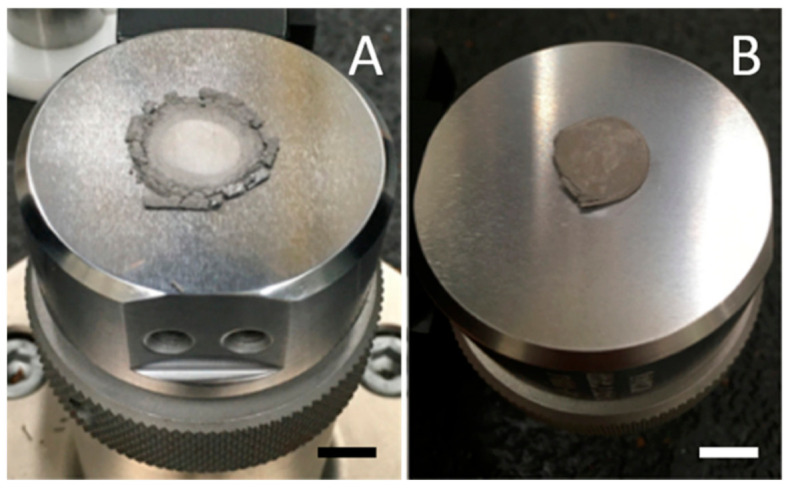
Contrasting control and HEAT samples under compression. (**A**) The control sample prepared at 850 °C is brittle and pulverized at ~1000 Atm pressure. (**B**) The HEAT-treated sample, 750 °C, is only modestly plastically deformed even at 5000 Atm pressure.

**Figure 8 materials-13-05008-f008:**
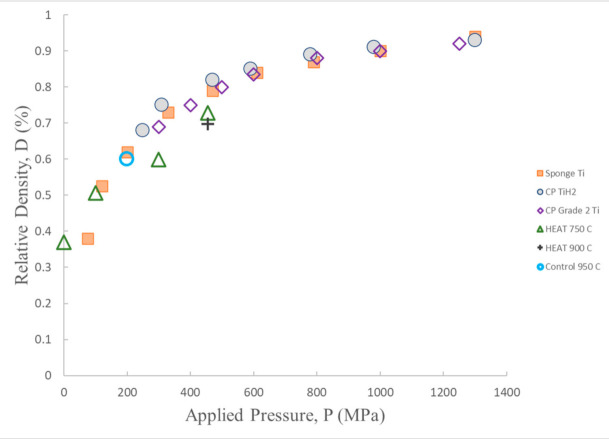
Cold compression—density following compression for HEAT samples treated at different temperatures, and one (macro cracked) control sample (950 °C) are shown. Also shown data from prior titanium cold compression studies of Ti and TiH_2_ powders [29]. The HEAT sample data follow the same trend as earlier data. Apparent increased hardness of HEAT samples may be an artifact, see text. After Ref [28].

**Table 1 materials-13-05008-t001:** Test description—pure Titanium.

Test	Firing Temp (°C)	Flow	Key Visual Observations—Raw	Key Visual Observations— Post-(5000 Atm) Compression Test
1	650	Ar/H_2_	Solid	Metallic Bonded Solid/modest shape change/SBB
2	650	Ar	Powder	-
3	650	Ar	Powder	-
4	650	Ar/H_2_	Solid	Metallic Bonded Solid/modest shape change (5000 Atm)/SBB
5	550	Ar/H_2_	Unstable solid	-
6	750	Ar/H_2_	Solid	Metallic Bonded Solid/modest shape change(5000 Atm)/SBB
7	750	Ar	Powder	Only powder after compression (5000 Atm).
8	850	Ar/H_2_	Solid	Metallic Bonded Solid/modest shape change (5000 Atm)/SBB
9	950	Ar/H_2_	Solid	Metallic Bonded Solid/modest shape change(5000 Atm)/SBB
10	850	Ar	Solid	Only powder remains
11	850	Ar	Solid	Retest of 10, only powder remained (confirmation)
12	750	Ar/H_2_	Solid	Metallic Bonded Solid/modest shape change (1000 Atm)/SBB
13	750	Ar/H_2_	Solid	Metallic Bonded Solid/modest shape change (3000 Atm)/SBB
14	900	Ar/H_2_	Solid	Metallic Bonded Solid/modest shape change(5000 Atm)/SBB
15	950	Ar	Solid	Cracked Solid (1800 Atm)

**Table 2 materials-13-05008-t002:** Density of samples fired using Ar/H_2_ gas.

Test	Firing Temp (°C)	Density (g/cm^3^)	Fraction Solid
1	650	1.44	0.32
6	750	1.786	0.396
8	850	1.919	0.426
12	750	1.659	0.368
13	750	1.652	0.367
14	900	1.668	0.370

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
