# Peer review of "Creating Strong Titanium/Titanium Hydride Brown Bodies at Ambient Pressure and Moderate Temperatures"

_materials, 2020, doi:10.3390/ma13215008_

Round 1
Reviewer 1 Report
Good WORK ABOUT HEAT process; specifically treating a mold filled with titanium 395 particles to above 650 ºC in a flowing ambient pressure gas containing hydrogen, creates a metallic 396 bonded brown body on the order of 40% metal density. Good English and grammar, very clerar.
Only some details to improve:
Ref 33 was very discussed among some researchers. Perhaps you can replace it.
On the other hand there are a lot of references from your group, even from the 80s. I can´t help but wonder if it would be nice in the journal, 5-6 can be logic, showing recent works.
Here in Britain the work about titanium is in line with recent advances in Ultrafan demonstrator. In our references we have all works by Beranoagirre regarding gamma Ti alloys. You can link your approach about the bonding with later works about secondary process, such as Beranoagirre (milling, turning, others).
A model 5892 Instron tester (Instron, Norwood, MA USA) was used in 169 ambient temperature compression mode (max capacity 100 kN) to determine density vs compressive 170 pressure behavior: please check that machine deformation was corrected in the calculations.
So you see, paper is quite interesting, but above formal aspects can be improved: it is not necessary to include all your work can link with people working on TRL level higher (Beranoagirre). By the way, titanium is a hot topic in UK all the time.
Author Response
Response to Reviewer 1:
We appreciate your time and effort and have done our best to respond to concerns. In our opinion the paper is now improved!
Reviewer 1 appears quite ‘enthusiastic’ about the work. The primary concern expressed is that there is insufficient attention given to the potential value of replacing some machining of titanium and titanium alloys, particularly Ti-Al alloys with the HEAT process. In order to address this issue, a significant modification was made in the Conclusions. The relevant section (paragraph 2 of ‘Conclusions’) now reads:
“It must be noted that finding a low cost, relatively rapid alternative to standard ‘subtractive’ manufacturing is particularly valuable for titanium and titanium alloys. Not only is titanium more expensive than other metals of construction, it is also clear cutting, drilling, and other standard metal shaping processes are particularly difficult for titanium and its alloys (52-53). It is argued below that HEAT is arguable the most viable alternative to standard titanium manufacturing. To wit: The HEAT process may have advantages relative to current commercial additive manufacturing processes, particularly as it creates brown Ti bodies at 650 °C and ambient pressure, with properties…”
Regarding Instron- Machine is under contract for repair and regular calibration. Used by many at NPS and appears accurate to all who use it.
Reviewer 2 Report
Please see the attached file.

Author Response
Response to Reviewer 2-
We appreciate your time and effort, clearly a very thorough job, and have done our best to respond to concerns. In our opinion the paper is now improved!
Comment 1-
P2, L55: lead to leads to …
In introduction, the authors need to summarize the differences between this research and existing studies?
Response- Our understanding of the issue is that P2, L55 is unclear. We changed the text to clarify:
“The proposed model of sintering in the HEAT process is that it is a form of Ostwald Ripening (OR) and that Ostwald Ripening does not occur in the absence of hydrogen gas. Thus, the nature of titanium sintering observed in the present study is unique. ”
Comment 2-
Section 2.1. How the titanium metal particles were added? It should be explained in detail.
Response- We modified the description of the particle addition as follows:
“ Second, titanium metal particles were added, simply by pouring particles from the source and gently brushing to create a flat top surface, until the mold was at approximately half capacity.”
It is a VERY SIMPLE process!
Comment 3-
In many places, “C” _indicates degree Celsius, it should be “°C” _
Response- Corrected.
Comment 4-
Fig. 3 is not clear.
Response- We reworded the caption. We changed the axes. Hopefully, all ambiguity has been removed:
Figure 3. XRD of Fired Samples- Firing temperature and gas phase shown. A) Commercial Ti powder. B-C) Pure titanium samples fired in Ar. D) Commercial TiH2 powder. E-F) Samples fired in forming gas. At temperature marked. These clearly contain both Ti metal and TiH2. The diffraction lines marked with the line segments are clearly TiH2, per D. All unmarked lines are Ti metal lines, per A.
Comment 5-
“TABLE 1” in L134 and “Table I” in L138.
Response- Now ‘Table’ throughout.
Comment 6-
L147: Lindburgh-Blue M 24” single zone… --> Lindberg/Blue, 24 inches
Response- Corrected.
Comment 7-
References section: Titles of some references were missing such as [27], [30],… Please check
Response- Titles of articles added.
Reviewer 3 Report
Dear Authors,
Thank you for your contribution on strong titanium/titanium hydride brown bodies.
I have the following issues:
1. p. 6/line 148: to me, it was not clear in which condition the tube was quickly placed inside the furnace. Is the tube sealed? If not, what about oxygen contamination?Same question about cooling after removal from furnace. A sketch of the experimental procedure would be really helpful.
2. I propose to put table 1 after the description of firing, since it contains the firing temperature
3. p6/l 168: how cylindric are your samples? If you determine density from volume & masse, your volume determination needs to be really (!) precise (See fig. 2B - to me, volume determination appears tough by measurement). Please explain your approach.
4. Fig. 3: very low image quality, axes cannot be read
Author Response
Response to Reviewer 3:
We appreciate your time and effort, and note that you have performed an unusually thorough review! We have done our best to respond to concerns. In our opinion the paper is now improved!
Comment 1. p. 6/line 148: to me, it was not clear in which condition the tube was quickly placed inside the furnace. Is the tube sealed? If not, what about oxygen contamination?Same question about cooling after removal from furnace. A sketch of the experimental procedure would be really helpful.
Response: The description was re-written:
“iv) Tube, sealed with no change in gas flow (per iii) quickly placed inside a furnace (Lindburgh/Blue M, 24” single zone)…”
And we added this remark at the end of the paragraph:
“(A picture of the furnace and the tube is available in Ref 16.)”
Comment 2. I propose to put table 1 after the description of firing, since it contains the firing temperature
Response 2: Not clear that will be an improvement in the flow of the narrative.
Comment 3. p6/l 168: how cylindric are your samples? If you determine density from volume & masse, your volume determination needs to be really (!) precise (See fig. 2B - to me, volume determination appears tough by measurement). Please explain your approach.
Response 3: We employed the same method as used by others. We are quite certain that our volume determination is within a few percent of the actual ‘apparent’ volume.
From Section 2.2-“ iii) An Archimedes principle device, ‘Ohaus density kit’ (Ohaus, Persippany NJ, USA), was employed in an effort to determine density. The Archimedies method results were found to be unrepeatable and clearly suggested an exaggerated density. This is consistent with the general finding that the Archimedes method is only reliable for titanium samples with density acceding 95% of nominal solid density [22,28-30]. Thus, like others [28-30], for the low density samples made in this study, a digital micrometer with a sensitivity of 0.02 mm was employed. Per standard practice values reported are based on average five measurements of thickness and diameter .”
- Fig. 3: very low image quality, axes cannot be read
Response 4- Figure 3 axes are improved, and the caption re-written to eliminate ambiguity.